
**Storm Tide Amplification and Habitat Changes due to Urbanization of a Lagoonal Estuary**
Philip M. Orton[1], Eric W. Sanderson[2], Stefan A. Talke[3,4], Mario Giampieri[2,5], Kytt MacManus[6]
1   Stevens Institute of Technology, Department of Civil, Environmental and Ocean
6          Engineering, Davidson Laboratory, Castle Point on Hudson, Hoboken, NJ 07030 USA
2   Wildlife Conservation Society, 2300 Southern Blvd., Bronx, NY 10460, USA
3   Portland State University, Department of Civil and Environmental Engineering, Post
9          Office Box 751, Portland, OR 97207, USA
4   Now at: California Polytechnic State University, Department of Civil and Environmental
11         Engineering, San Luis Obispo, CA 93407, USA
5   Now at: Massachusetts Institute of Technology, School of Architecture + Planning,
13         Department of Urban Studies and Planning, 77 Massachusetts Avenue, Cambridge, MA
14         02139, USA
6   Columbia University, Center for International Earth Science Information Networks, PO
16         Box 1000, 61 Route 9W, Palisades, NY 10964, USA

Correspondence:  Philip Orton (philip.orton@stevens.edu)




**Abstract**
In recent centuries, human activities have greatly modified the geomorphology of coastal
regions. However, studies of historical and possible future changes in coastal flood extremes
typically ignore the influence of geomorphic change. Here, we quantify the influence of 20th
Century manmade changes to Jamaica Bay, New York City, on present-day storm tides. We
develop and validate a hydrodynamic model for the 1870s, based on detailed maps of
bathymetry, seabed characteristics, topography, and tide observations, for use alongside a
present-day model. Predominantly through dredging, landfill, and inlet stabilization, the
average water depth of the bay increased from 1.7 to 4.5 m, tidal surface area decreased from
92 to 72 km$^2$, and the inlet minimum cross-sectional area expanded from 4800 to 8900 m$^2$.
Total (freshwater plus salt) marsh habitat area has declined from 61 to 15 km$^2$ and intertidal
unvegetated habitat area from 17 to 4.6 km$^2$. A probabilistic flood hazard assessment with
simulations of 144 storm events reveals that the landscape changes caused an increase of 0.28
m (12%) in the 100-year storm tide, even larger than the influence of global sea level rise of
about 0.23 m since the 1870s. Specific anthropogenic changes to estuary depth, area and inlet
depth and width are shown through targeted modeling and dynamics-based considerations to
be the most important drivers of increasing storm tides.
**Keywords:** Estuary; storm surge; geomorphology; habitat; hazard assessment; dredging;
landfill, Jamaica Bay, New York
**1. Introduction**
The characteristics of storm tides and the probability of flooding depend on both far-field
forcing (meteorological, tidal) and on local characteristics (bathymetry, bottom roughness,
floodplain size). Therefore, changes to local mean sea level, shipping channel depths, wetland
land cover, and storm intensities, sizes, speeds, and tracks can all potentially alter system
response and flood probabilities. Recent non-stationary, probabilistic hazard assessments have
demonstrated spatially coherent variability in common storm tides (Marcos et al., 2015), as well
as extreme storm tides (Wahl and Chambers, 2016), and have begun revealing the climate
modes (e.g., NAO and ENSO index) that modulate storm tides in some regions. Similarly, long
term cycles in astronomic forcing (e.g., the 18.6-year nodal cycle) affect both nuisance flooding
(Ray and Foster, 2016) and the probability of high impact events (Talke et al., 2018). In some
estuaries, such as Boston Harbor, flood hazard remains statistically stationary after accounting
for sea-level rise and tidal variability (Talke et al., 2018). In others, flood hazard is non-
stationary. For example, a recent study of New York Harbor (NYH) showed an increase in the
10-year storm tide of 0.28 m since the mid-1800s in addition to the local relative sea level rise
of 0.44 m (Talke et al., 2014).

Climatic and astronomical variability in hydrodynamic forcing coincides with several centuries
of human-induced geomorphic change to estuaries and harbors (e.g., Sanderson, 2009;
Grossinger, 2001; Talke et al., 2018; Jaffe et al., 1998). Wetlands have been reclaimed; in NYH, a


typical case, approximately 80% of pre-development wetlands have been lost (USACE, 2009).
Harbors and estuaries have been deepened, with the controlling depth of channels often
doubled or even tripled (Orton et al., 2015; Familikhalili and Talke, 2016; Ralston et al., 2019;
Helaire et al., in press; Chant et al., 2011). Coastal boundaries have been hardened and raised,
preventing overland flooding except in extreme cases. Natural wave breakers have been
destroyed, including oyster reefs that may have once reduced coastal wave energy in New
York's outer harbor by between 30 and 200% (Brandon et al., 2016).
The sum effect of changing bathymetry is an altered hydrodynamic regime, with effects on
astronomical tides, storm surges, and morphodynamic feedbacks (e.g., de Jonge et al., 2014;
Chernetsky et al., 2010; Talke and Jay, 2020). A study of the Cape Fear Estuary showed that tide
range had doubled since the 1880s in Wilmington, NC, due to a doubling of the shipping
channel. Moreover, idealized modeling showed that a ~0.5 to 2 m storm surge increase at
Wilmington across a variety of hurricane intensities (Familkhalili and Talke, 2016). Model
simulations of Hurricane Katrina's flooding with present-day versus estimated historical
conditions (ca. 1900) suggest that wetland loss exacerbated flooding well beyond the influence
of sea level rise (Irish et al., 2014). Within the Hudson River estuary, Ralston et al. (2019)
showed that a doubling of channel depth near Albany (NY) more than doubled tide range and
increased the magnitude of storm surge compared to 19[th] century conditions. Within New York
Harbor, deepening of the inlet produced a smaller shift in the lunar semidiurnal tidal
constituent amplitude of 7% at The Battery (Ralston et al., 2019). Within nearby Newark Bay
and the Passaic River, tides have been amplified by ~10% over the past century, reflecting a
change in the controlling channel depth at some locations from ~3 to 15 m (Chant et al., 2011).
In parts of Jamaica Bay, another sub-embayment of New York Harbor, tide range changes are
much larger and have grown by 41%, from 1.16 m in 1899 to 1.64 m in 2007 (Swanson and
Wilson, 2008). Numerical experiments within Jamaica Bay suggest that individual storm tide
events such as Hurricane Sandy are quite sensitive to depth modifications (Orton et al., 2015).
However, the implications of historical channel deepening and land cover changes on flood
hazard have not yet been quantified through a probabilistic assessment.
In this contribution, we investigate the influence of extreme changes in bathymetry and
wetland cover on storm tide hazard. Jamaica Bay, New York, was a back-bay lagoonal system
that was converted to a deepwater port (Sanderson, 2016; Swanson and Wilson, 2008; Seavitt
et al., 2015; Swanson et al., 2016). Although the system's morphology was evolving in the 18[th]
and 19[th] centuries and possibly earlier, the most dramatic alterations occurred in the early 20[th]
century (Black, 1981). The Jamaica Bay Improvement Commission (1907) proposed to
reconfigure the bay into a port (**Figs. 1-2**), and The River and Harbor Acts of 1910 and 1925 set
in motion a plan to reconfigure the entrance channel to a depth of at least 9 m and width of
450 m, protected by jetties. Groins were placed along the seaward-side of the Rockaway
Peninsula (labeled in Fig. 1 as "Rockaway Beach") and a jetty constructed at the tip to stabilize
the barrier island (Hess and Harris, 1987). The bay's perimeter channels were extensively
dredged for several decades, and dredged sediments were used for landfill development over
the fringe wetlands surrounding the bay, creating neighborhoods and the Floyd Bennett Field
airport (Black, 1981). At mid-century, additional dredging and landfill occurred at the




northeastern end of the bay, for creation of John F. Kennedy (JFK) International Airport, leaving
"borrow" pits that today are up to 15 m deep. As the 20th Century progressed, the port was
never realized, and the primary port for the region ended up across New York Harbor in Newark
Bay.
Here we present a quantitative assessment of Jamaica Bay landscape changes and use
retrospective modeling to estimate the impacts on storm tides and flooding. A detailed
hydrodynamic model of the 1870s was developed based on maps of bathymetry and seabed
characteristics, for use alongside an existing present-day model. Modeling of 144 storm tide
events for both the 1870s landscape and the present-day landscape is used to develop a
probabilistic flood hazard assessment. We show that manmade geomorphic changes in Jamaica
Bay have produced an important and heretofore under-appreciated and unquantified increase
in storm tides. Given the environmental and societal value of the Jamaica Bay wildlife refuge,
JFK Airport, the Gateway National Recreation Area, several city and state parks, and the lives of
the hundreds of thousands of people in flood zones around the bay, our results have
implications for the future management of the system.

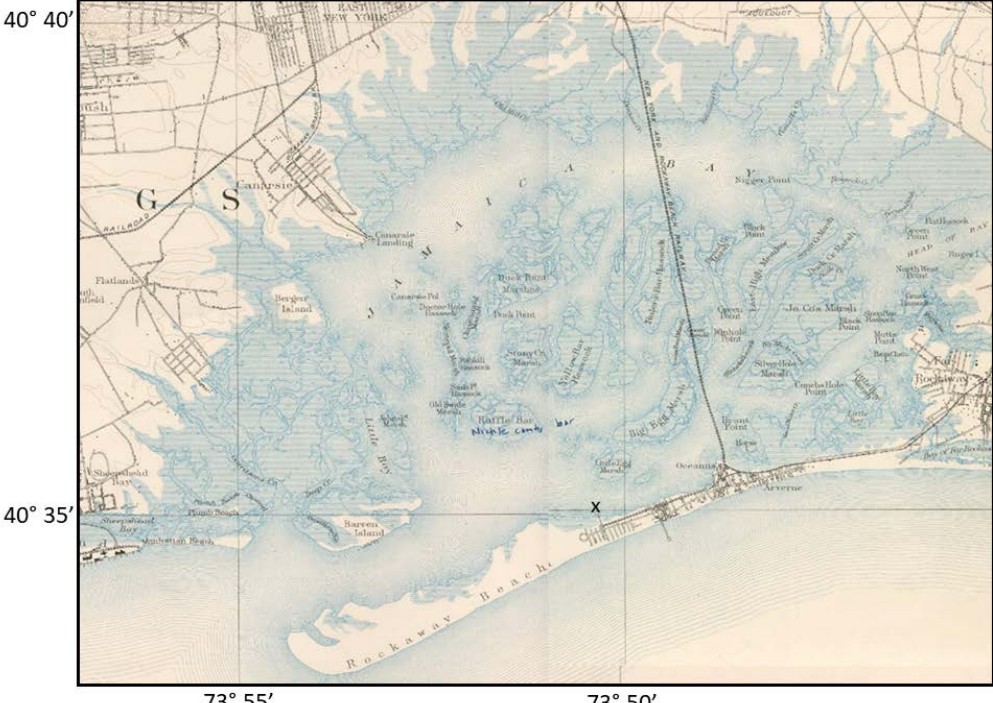

**Figure 1**: An 1888-1889 survey map of Jamaica Bay, in southeast New York City, portraying the
morphology and marsh cover (blue hatching). The map is excerpted from Powell (1891) and the
"x" marks the Holland House pier tide measurement location.

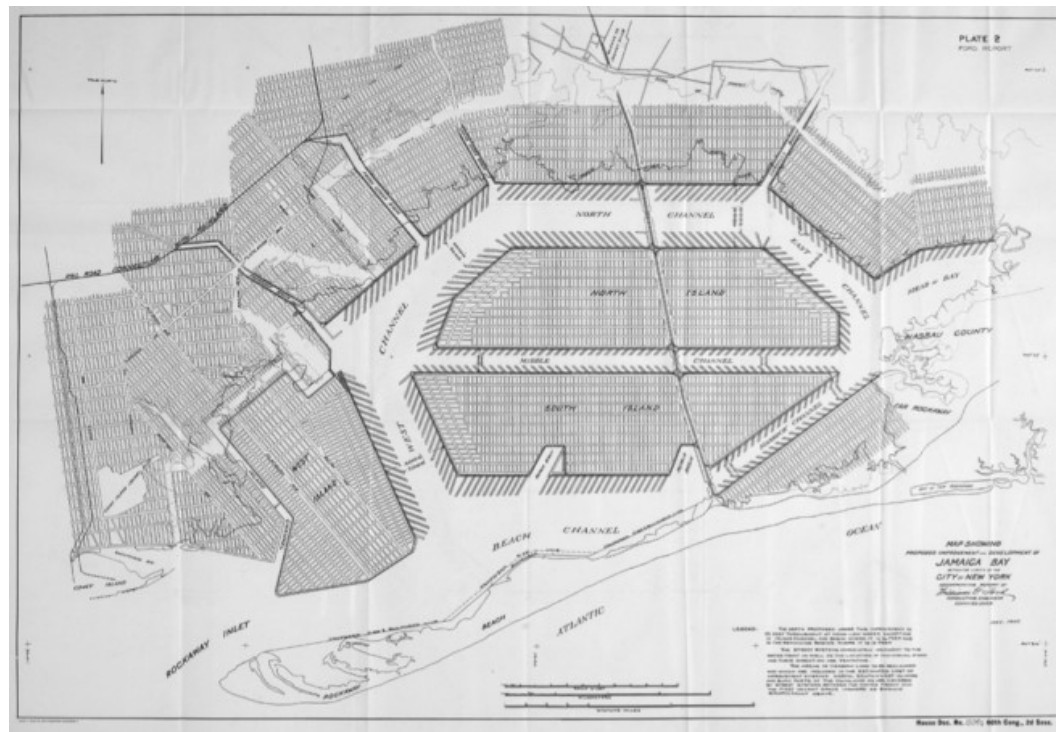

3
**Figure 2**: Plan for converting Jamaica Bay into a port (Jamaica Bay Improvement Commission, 1907)

## 2. Methods

To evaluate how and why flood hazard has changed due to landscape changes in Jamaica Bay (see Results), we applied a quantitative approach—the use of numerical models to produce a probabilistic hazard assessment (e.g., Orton et al., 2016)—to both the historical (1870s era) and modern bathymetries and landscapes of Jamaica Bay. Below, we describe our landscape reconstruction (2.1), our modeling approach (2.2), and our hazard assessment methodology (2.3).

### 2.1 Jamaica Bay landscape reconstructions

Although maps and charts of the Jamaica Bay landscape extend back to the 17th century (Sanderson, 2016), the first thorough bathymetric and topographic maps were made by the US Coast Survey between the 1840s and 1870s. The first tidal measurements also date from this period (e.g., Talke and Jay, 2013). Because the 1870s time period pre-dates most channel deepening, this period constitutes a good proxy for conditions prior to major 20th century anthropogenic modifications.




To develop numerical models of the "present-day" and 1870s conditions, we first created
digital elevation models and land cover maps at 30 m resolution. The domain extends eastward
and northward to land up to 6 m navd88 elevation, and extends westward past Coney Island.
The landscape reconstruction from the 1870s forms a way-point between the pre-European
landscape of c. 1609 and modern conditions (Sanderson, 2016). Since no bathymetric data are
available from before the 19[th] century, comparisons between the 1600s and 1800s are
qualitative (See **Sect. 4.3).**
*2.1.1 Present-day landscape*
The present-day digital elevation model is based (by order of preference) on United States
Geological Survey (USGS) bathymetric/topographic data collected by LIDAR in 2013–2014,
slightly older data collected in 2007-2008 by Flood (2011), and older National Oceanic and
Atmospheric Administration bathymetric survey data for a few remaining small areas of the
Bay. The LIDAR data cover dry land, marsh islands, and shallow waters (shallower than
approximately 2 m) and the Flood (2011) data cover the navigation channel and other deep-
water regions. Bare-earth land elevations in populated areas are based on 2010 New York City
LiDAR data. Present-day land cover data for the Jamaica Bay watershed at 30 m resolution are
from the 2011 National Land Cover Dataset (NLCD), as described in Homer et al. (2015).
*2.1.2 Historical landscape data*
Bathymetric and benthic character data for the 1870s model are from a pair of H-sheets from
1877 and 1878 for Jamaica Bay: Maynard (1877) and Moore (1878). The Maynard (1877) survey
was drawn at 1:5,000 scale, while the Moore (1878) survey was drawn at a scale of 1:10,000.
Both show grids of depth surveys, with parallel lines approximately 100 m apart, and with
sounding data approximately every 20 m (**Fig. 3**). Moore (1878) includes depth contour lines
that mark out channels between the marshy islands and other underwater features. While
earlier H-sheets depicted the bathymetry of Rockaway Inlet and Broad Channel, the Maynard
(1877) and Moore (1878) manuscript maps are the first to depict the bathymetry of the entirety
of Jamaica Bay. Approximately 20,000 individual sounding points were digitized to describe the
interior of the bay. Raw data were corrected for tidal stage and reduced to the Mean Low
Water datum, based on local tide gauge measurements. Since we have recovered and digitized
these hand-collected tide records from the US National Archives (see e.g., Talke and Jay, 2017),
we are able to validate our model results for the historical model against contemporary 1870s
data (see **Sect. 2.2**).
Topographic and land-cover data were digitized and synthesized from T-sheets and other
surveys drawn by Bien and Vermule (1891a), Bache (1882), Bien and Vermule (1891b), Dorr
(1860), Gilbert (1855), Gilbert (1856a), Gilbert (1856b), Gilbert and Sullivan (1857), Jenkins
(1837a, b), Powell (1891), and Wilson (1897). Historic maps and charts were georeferenced
using a first order rectification to the modern city grid with less than 50 m root mean square
error, using control points located at road intersections, buildings, railroads, or other features
that are present historically and can be located today. To reduce to a common datum and
assess temporal evolution, we tracked the datum of each map or chart and the publication
date.

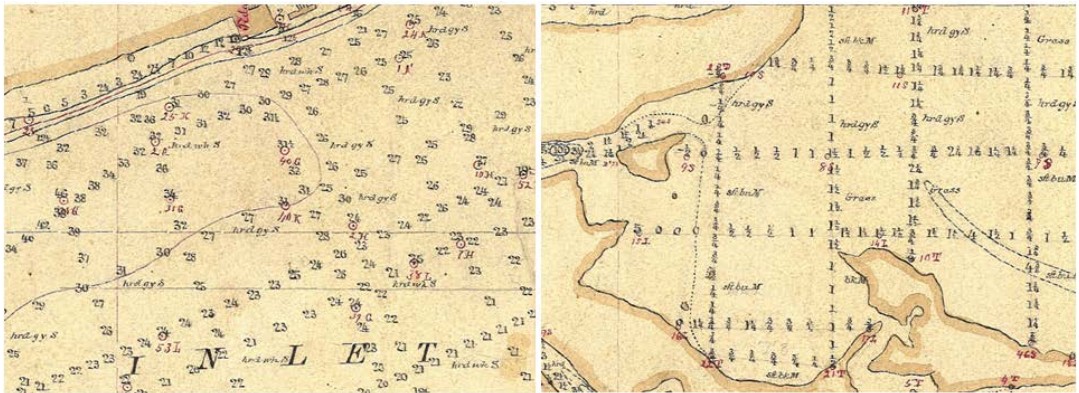

**Figure 3**:  Detail view of two portions of the 1877 survey dataset (left) at Rockaway Inlet (at bottom left of **Figs. 1-2**) and (right) a shallow bay area with mud, sand and grass areas (Maynard, 1877). Shown are measured depths (in feet) and bottom characterization notes (e.g. "sft"  for soft, "hrd" for hard, "gy" for gray, "S" for sand, "M" for mud, and "Grass" likely for eelgrass beds), with typical spacing of 100-150 m. The mapped area on the right is now covered by fill and a former airport, Floyd Bennett Field.

Because historical surveys usually neglected intertidal areas, we use inferential techniques to approximate the historical elevations within this region, using known plant-cover data. Specifically, the present-day vertical zonation of salt marshes around New York City was used to approximate the historical elevation of marshes. The seaward extent of salt marsh was assumed to represent the mean sea level (the lower edge of the low salt marsh; Edinger, 2014), while the landward edge was assumed to represent the extent of high tide flooding (the upper edge of the high salt marsh; Edinger, 2014). Locations where maps showed a contour between low and high salt marsh were assigned an elevation equal to mean high water.

Vertical datum adjustments were made by relating the topographic zero of each map and chart to the relative sea level reconstruction (RSL) provided by Kemp & Horton (2013). They studied foraminiferal assemblages over the past two centuries from salt marsh sediment in nearby Barnegat Bay, New Jersey. Their results were used to identify RSL in the southern coastal New York City at the time the map or chart represents. To estimate the NAVD88 elevation of the topographic zero for the map, we noted that the Kemp & Horton (2013) study places the 0 level of their RSL reconstruction at 0.10 meter above mean sea level in Barnegat Bay, which was converted to NAVD88 using NOAA Tides & Currents adjustment values for Barnegat Inlet (Station 8533615).

Raster digital elevation models (DEM) were created in ArcGIS 10.3 with the "Topo to Raster" interpolation method to create hydrologically correct DEMs (ESRI, 2016). In addition to contour





line and point elevation data, historical stream and pond data were also added. To preserve the
winding characteristics of marsh creeks during the interpolation, creek beds were converted to
point features and their elevation was set at the Mean Low Water datum of the appropriate
date.
***2.2 Flood and tide modeling and validation***
A hydrodynamic model was applied to the historical and modern "landscapes" (land surface
elevation and roughness) and used to simulate an ensemble of storm tides following the
methodology of Orton et al. (2016b). The hydrodynamic model sECOM provides validated and
accurate (Georgas et al., 2014; Orton et al., 2012) ensemble 3D storm tide predictions as part of
the NY Harbor Observation and Prediction System (NYHOPS; Georgas and Blumberg, 2010) and
the Stevens Flood Advisory System (Jordi et al., 2018). The Jamaica Bay model grid was a 30 by
30 meter, square-cell grid (Orton et al., 2015). This grid was doubly-nested inside two larger
model domains that represent (1) the regional coastal ocean and estuaries from Maryland to
Cape Cod, and (2) the Atlantic Ocean from Cape Hatteras to Nova Scotia (Orton et al., 2016b).
Storm meteorological forcing for the regional and large-scale grids was spatially and temporally
variable, and is described in Orton et al. (2016b) and the next section.
Simplifying assumptions are used for the model simulations on the Jamaica Bay grid for
computational efficiency in simulating a large number of storms. While the regional coastal and
estuary modeling used 3D simulations, the model's two-dimensional (2D) mode was used for
Jamaica Bay (e.g., Orton et al., 2015). This is a common practice in estuary storm tide modeling
(Familkhalili and Talke, 2016; Kennedy et al., 2011). While stratification can have a small
influence on storm tides in stratified estuaries (Orton et al., 2012), Jamaica Bay has limited
freshwater input and weak stratification (Marsooli et al., 2018). A wave model is not coupled
with the hydrodynamic model, for computational efficiency and because our focus here is on
storm tides and "still water" elevations. The broad shallow continental shelf at the Apex of New
York Bight leads to relatively small impacts of waves on estuary storm tide temporal maxima
(e.g. due to wave set-up; Marsooli and Lin, 2018; Lin et al., 2012). Lastly, the time-varying
meteorological forcing was assumed spatially constant on the Jamaica Bay grid, because the
bay is only ~10 km wide.
The gridded land elevation and land cover type datasets for the 1870s and present-day were
interpolated onto the model grid to create land elevation and Mannings-n roughness model
input files. The 30-meter resolution modeling does not resolve fine-scale features such as
elevated seawalls, though they are rare in this area. In 2D tide and storm surge modeling
studies, a common simplified approach (Irish et al., 2014; Mattocks and Forbes, 2008; Szpilka et
al., 2016) to representing the effects of wetlands and other natural features is to treat them as
enhanced landscape roughness features, through a variable called Mannings-*n*. Reasonable
estimates for Mannings-*n* values are 0.045 for intertidal wetlands and eelgrass (Zostera Marina)
beds, 0.020 for unvegetated continental shelf and estuary substrate, and 0.10 and 0.13 for
medium and high intensity developed land, respectively (Mattocks and Forbes, 2008).  This
approach has previously been applied to Jamaica Bay (Orton et al., 2015).





Depending on purpose, different mean sea-levels were used in the study. To determine habitat
and tidal datum changes, we run tide-only simulations using the mean sea level that existed for
a given landscape year. Storm simulations for both the modern and historic (1870s) period use
2015 mean sea level, to quantify the effect of landscape change on flood hazard and isolate this
process from the effect of sea level change. Mean sea level for the 1870s was -0.28 m (Kemp
and Horton, 2013) and in 2015 was +0.09 (based on smoothed recent trends), both relative to
the 1983-2001 MSL datum at The Battery (NOAA station 8518750). These values are -0.37 and
0.00 m NAVD88, respectively, based on conversions for the Jamaica Bay Inwood tide gauge
(USGS station 01311850). An elevated (or reduced) mean sea level was imposed as a constant
offset to a given simulation's offshore elevation boundary conditions at the edge of the Jamaica
Bay grid. This is a reasonable simplification here because recent work showed virtually no
change to tides at nearby Sandy Hook (NOAA station 8531680) when there is sea level rise
(below a 1% change to tide range per meter of sea level rise Kemp et al., 2017).
Tide-only simulations for 1878 were run for a 40-day period that overlapped with water level
observations made from 13 August 1878 through 21 September 1878 at a pier on the north side
of the Rockaway Peninsula (**Fig. 1**). The tide simulation for the present-day covers a 35-day
period from 1 August 2015 through 5 September 2015. Since wind-forcing during the late
summer is typically weak, these tide-only simulations are useful for direct validation of the
model.
Model validations were performed for the 1870s era model, and the present-day model was
previously validated (Orton et al., 2015). The prior storm validation of the present-day model
for Hurricane Sandy showed a time series RMSE of 20 cm and high water mark RMSE of 19 cm
(Orton et al. 2015). The tidal validations here use summertime periods without strong wind
influences, and modeled time series were compared to observations for both 1878 and 2015
using RMS error and the Willmott skill (e.g., Warner et al., 2005). The 2015 period included
7920 samples taken at 6-minute intervals over a 33-day period at the Inwood USGS gauge
station. The 1878 period included only daytime measurements, with 2438 samples taken at 10-
minute intervals over a 37-day period at the Holland House pier on the north side of Rockaway
Peninsula. The mean error is subtracted before computing statistics to account for possible
remote sea level anomalies or steric sea level variations, and because the 1878 tide staff datum
is poorly known. The results for the tide modeling time series validation for 1878 were 0.09 m
RMS error and 0.991 skill, while the results for the 2015 period were 0.09 m RMS error and
0.989 skill.
Historic and modern tidal datums, tidally wetted area and intertidal zones were assessed by the
following methodology. First, simulated water levels after a 2 day spin-up period were
harmonically analyzed (Pawlowicz et al., 2002) at historic gauge locations. A year-long synthetic
tide time series was then produced, using appropriate nodal corrections, and once and twice-
daily water level minima and maxima were compiled and averaged to compute tidal datums
such as MLLW and MHHW. The tidally-wetted area was then defined as the area wetted at high
tide in Jamaica Bay after MHHW conditions at Rockaway Inlet. The intertidal area is similarly


defined as the difference between the tidally-wetted area and the area flooded at the low tide
occurring after a predicted MLLW tide at Rockaway Inlet.
***2.3 Probabilistic flood hazard assessment***
A probabilistic flood hazard assessment was used to quantify the annual probabilities of
exceedance (or inversely, the return periods) for any given storm tide. We applied the storm set
and statistical framework utilized by Orton et al. (2016b), which employed a joint probability
method of flood hazard assessment that is an ensemble simulation of a diverse set of possible
storms (the storm climatology) including both synthetic tropical cyclones (TCs; e.g. hurricanes)
and historical extratropical cyclones (ETCs; e.g. nor'easters). The synthetic TCs spanned all
combinations of a complete range of intensities (6 bins), sizes (3), speeds (3), landfall locations
(5) and angles (3), and each simulated TC had an estimated annual frequency of occurrence
based on an extensive simulation with a statistical-stochastic TC model (Hall and Yonekura,
2013). The wind and pressure meteorological forcing for ETCs was historical reanalysis data
from Oceanweather, Inc., whereas the forcing for TCs came from simplified parametric TC
models. The assessment methods were validated by comparison to historical data at multiple
levels of the study, demonstrating unbiased storm tide simulations and storm tide hazard
estimates (versus return period), relative to historical events (Orton et al., 2016b). Additional
details of the assessment, including historical data, validations, storm climatology development,
statistical analysis and uncertainty quantification are given in Orton et al. (2016b). The storm
tide modeling results from the larger-scale model grids in this prior study were applied as
offshore boundary conditions to the Jamaica Bay domain simulations for the present study.
Some simplifications of the application of the Orton et al. (2016b) flood hazard assessment to
our Jamaica Bay submodels are noted here. The prior flood hazard assessment included 1516
storm simulations (606 TCs, and 910 ETCs), but we use an abbreviated storm set to reduce the
computational expense. The abbreviated set of 80 ETCs includes all the same storm events, but
fewer random tide permutations for each storm. The abbreviated set of 64 TCs includes a range
of storm tide events from low to high magnitude (1.5 to 6.0 m). Model results for simulated TC
events at a given magnitude are then used as a proxy for all the events at that magnitude. A
statistical comparison of the abbreviated versus full storm set showed minor differences of less
than 5% across 5-year to 500-year storm return periods, validating our approach. The historic
and modern model landscapes are subjected to the same set of storms, and therefore any
differences in storm tide hazard reflect geomorphic changes rather than artifacts of the
simplified hazard assessment.
**3. Results**
Our digitization of the historical landscape shows that changes to Jamaica Bay land cover and
elevation since the 1870s are dramatic, with widespread urbanization of upland areas and
marshlands that once surrounded the bay. Maps of estimated Jamaica Bay area land cover for
the present-day and 1870s periods are shown in **Fig. 4**. The most dramatic land cover change is
from large areas of fringing wetlands (light blue) to urbanized areas (red), but also the center of




the bay has shifted from marshes to open waters (dark blue). Mapped land elevations
(topography, bathymetry) and Mannings-n roughness values are shown in **Fig. 5**. Obvious
geomorphic changes include a lengthening of Rockaway Peninsula and reconfiguration of the
inlet (bounded by red lines). The land roughness (Mannings-n) change reflects the widespread
change from marshes (light blue) to urbanized land (red) or open water (dark blue). These
changes in habitat type are quantified in **Sect. 3.1**, below.
Simulations suggest that the mean water depth in Jamaica Bay has increased by either 2.8 or
3.1 m, with the exact result dependent on how calculations are made.  If only wetted regions
are included in the average, water depth in Jamaica Bay increased from 1.7 m to 4.5 m between
the 1870s and 2015; of this change, 0.37 m can be attributed to sea-level rise.  If the entire
tidally-wetted bay area is used in an average (with dry grid cells included as zero depth),  a
historical and modern mean depth of 1.1 m  and 4.2 m is found. Our values are consistent with,
and improve upon, the approximate estimate of a historical change from 1 m to 5 m made by
Swanson et al. (1992). In conclusion, our results show a large historical change in bay-wide
mean depth, but slightly smaller than prior studies have suggested. A detailed analysis of areal
changes to various types of habitat is given in **Sect. 3.2**.

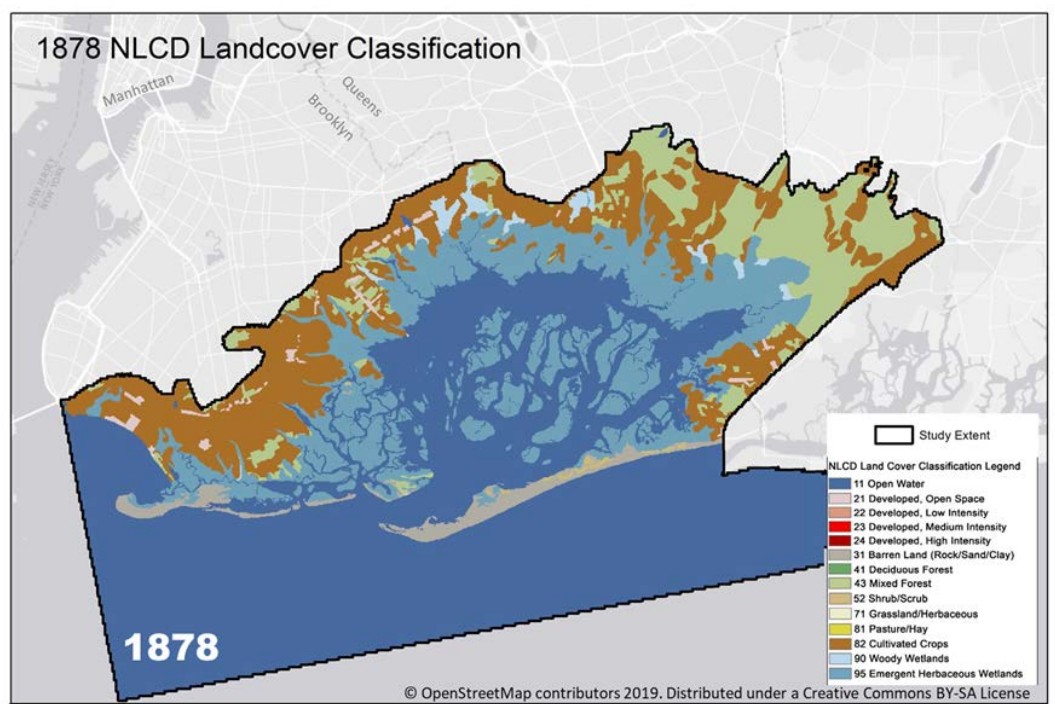

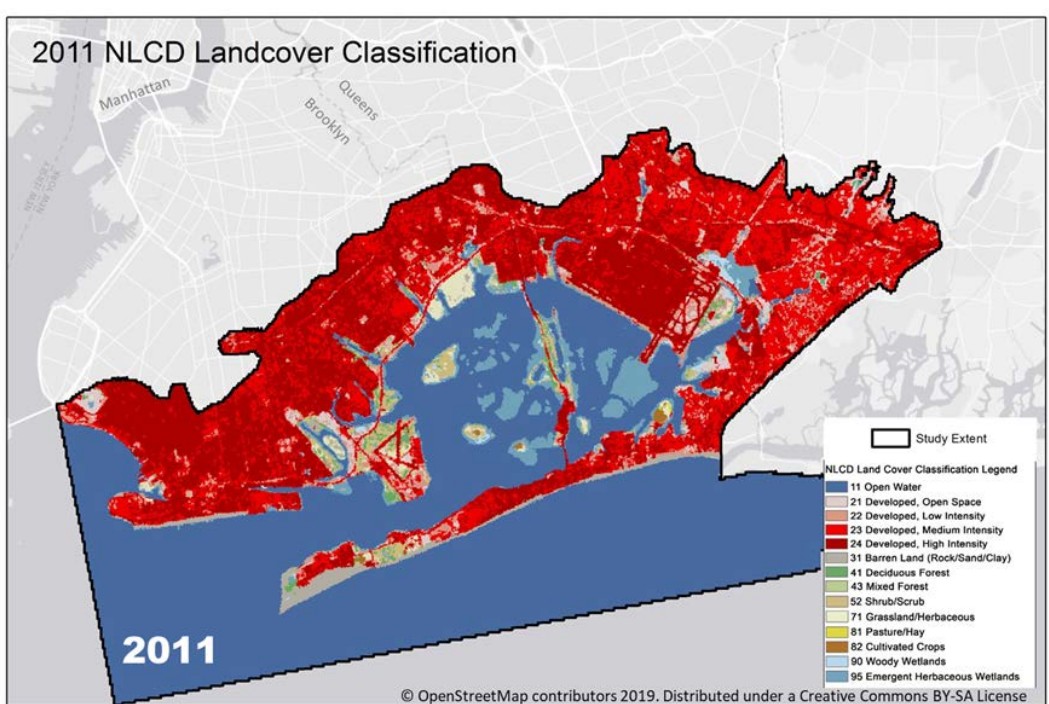

**Figure 4**: Land cover of the Jamaica Bay watershed (top) reconstructed for the 1870s, and (bottom) for present-day.

### 3.1 Habitat changes

The surface areas of many habitat types have changed dramatically since the 1870s, in spite of an only 23% reduction in interior bay area wetted by average daily high tides (**Table 1**). The reduction in total area is caused by the reclamation of fringing flood-plain and marshlands, but is partially offset by a growth of the bay westward due to an increase in inlet length.

Total marsh area has declined by 76%, eelgrass area by 100%, intertidal unvegetated area by 72%, and total intertidal area by 73%. The deepwater area (>4 m) has increased by 314% (or alternatively, the 1870s had 76% less deepwater area than the present). The estimates for wetland area and loss are nearly identical to the prior estimate of a loss of 75% from 64 km$^2$ to 16 km$^2$ (NYC-DEP, 2007), but here we provide greater context of changes to other habitat types. The habitat type changes are computed within the differing bay interiors for the 1870s and present-day, as enclosed by red line inlet boundaries shown in the top panels of **Fig. 5**.

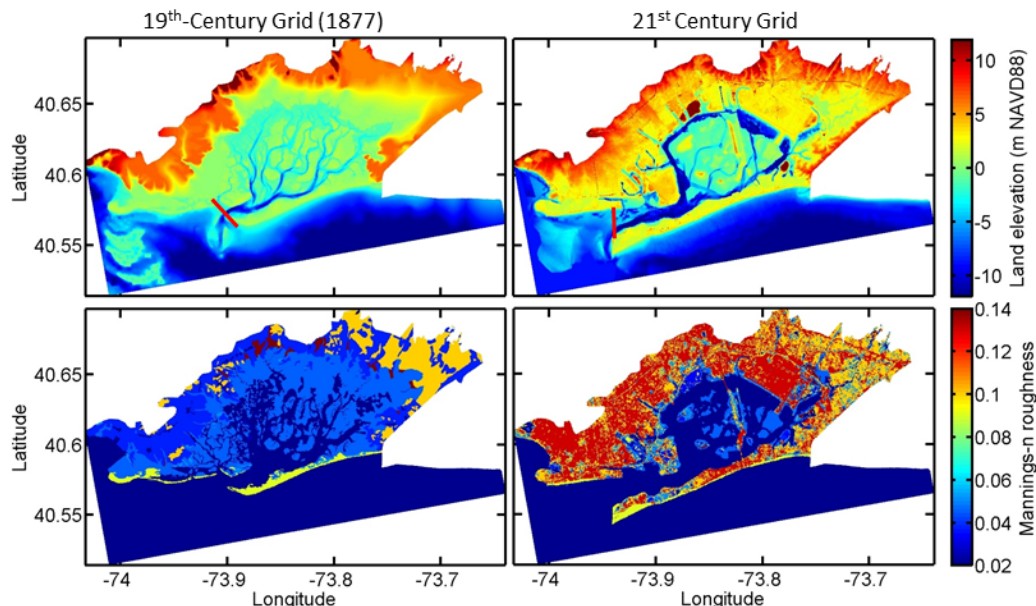

2
**Figure 5**: 1870s and early twenty-first century landscape data used as inputs to the
hydrodynamic model. On the **top** are land elevation maps, and on the **bottom** are land-cover
roughness (Mannings-n) maps. The **left column** shows the 1870s and the **right column** shows
the present-day landscape. Red lines delineate the inlet boundary for defining the interior of
the bay and tidal prism.

**Table 1:** Estuarine habitat types and their area for the 1870s and present-day

| Landscape | Total marsh area[a] (km²) | Eelgrass area (km²) | Intertidal-unvegetated (km²)[b] | Total intertidal area[b] (km²) | Deep area (>4 m) (km²) | Interior bay area[c] (km²) |
|---|---|---|---|---|---|---|
| Basis | map data | map data | map data, tide simulation | tide simulation | map data | map data |
| 1870s | 61.3 | 16.5 | 17.3 | 51.5 | 6.6 | 92.4 |
| Present-day | 14.9 | 0 | 4.9 | 14.0 | 27.7 | 71.5 |
| Change | -46.5 (-76%) | -16.5 (-100%) | -12.4 (-72%) | -37.5 (-73%) | 20.9 (314%) | -20.9 (-23%) |

a: Includes all saline marsh and freshwater marsh within the model domain, some not tidal
b: Intertidal area is the difference in area wetted by MHHW and MLLW, based on modeling
(**Sect. 2.2**)
c: Interior bay area is the wetted area at MHHW, based on modeling (**Sect. 2.2**)



*3.2 Storm tide changes*
The flood hazard assessment shows similar basic features as found in the prior study of New
York Harbor form which methods and offshore model boundary conditions were taken (Orton
et al., 2016b). The estimated storm tide for return periods below 30 years is determined
predominantly by the relatively frequent extratropical cyclones, and the curve (**Fig. 6**) has a
relatively small slope of storm tide with increasing return period. For return periods above 30
years, tropical cyclones become increasingly important and the slope abruptly increases at
about the 70-year return period.
The results reveal that storm tides are markedly larger on the present-day landscape than the
historical landscape across a wide range of return periods (**Fig. 6; Table 2**). Holding sea-level
constant at 2015 levels, the modern 10-year and 100-year storm tides of 2.02 and 2.66 m are
larger than historical simulations by 0.20 m and 0.28 m, respectively, at the eastern end of the
bay (Inwood).  By contrast, sea-level rise effects are small; when we simulate storms on the
1870s landscape with the 1870s sea level, the 100-year storm tide difference increases by 0.02
m to 0.30 m. The increase in storm-tides is attributable to decreased frictional effects, which
scale as $1/H$ (e.g., Friedrichs & Aubrey, 1994). Because the ~3 m increase in average depth
caused by landscape changes is much larger than the ~0.37 m increase in sea-level, landscape
changes dominate long term changes to flood hazard.

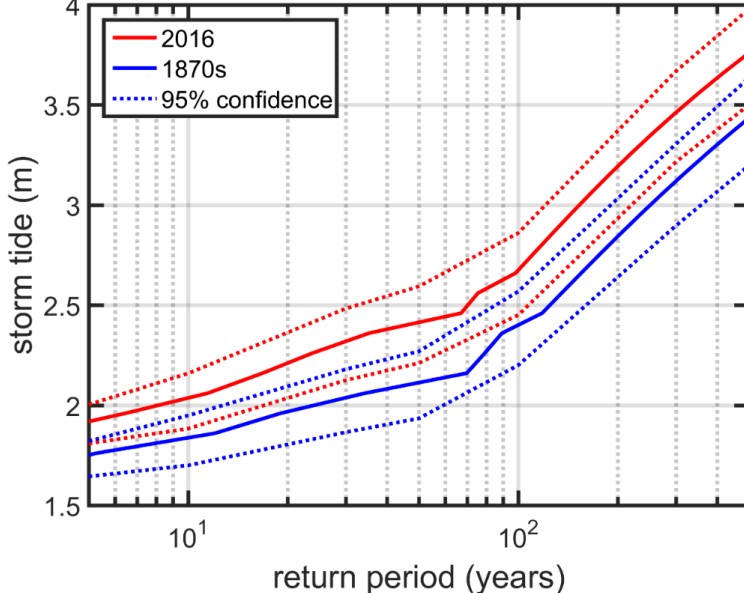

**Figure 6**: Storm tide exceedance curves for the 1870s and present-day Jamaica Bay landscapes,
for Inwood. Storm tide is the water level above mean sea level (MSL), and storms for both cases
were simulated with 2015 MSL.



**Table 2**: Storm tide elevation and flood area for 1870s versus present-day landscapes[a]

| Landscape | 10-year Storm tide (m) | 100-year Storm tide (m) | 10-year Flood area (km$^2$) | 100-year Flood area (km$^2$) |
|---|---|---|---|---|
| 1870s | 1.82 | 2.38 | 279 | 284 |
| Present-day | 2.02 | 2.66 | 226 | 243 |
| Change | 0.20 or 11% | 0.28 or 12% | -53 or -19% | -41 or -14% |

a: These are tallied across the entire model domain
Storm tides for the 1870s landscape are seen to clearly decrease with distance into the bay,
with the 100-year flood elevation declining from 2.54 m outside the inlet to 2.42 m in the
eastern part of the bay (**Fig. 7**). By contrast, present-day storm tides (and tides) amplify within
the bay, and therefore the 100-year flood hazard increases from 2.56 (outside the inlet) to 2.70
m (eastern bay).
Increases in storm tide magnitudes in the bay do not necessarily lead to increases in flooding
extent. While **Fig. 6** shows that storm tides are increased substantially by the landscape
changes from the 1870s to present, **Fig. 7** demonstrates that the flooded area has substantially
decreased for the 100-year flood. **Table 2** shows that the 100-year flood area decrease is 41
km$^2$ and the 10-year flood area decrease is 53 km$^2$ across the model domain (both including the
Coney Island and the Jamaica Bay areas). The simple explanation for this is that fringing
marshes across the region that were -0.25-0.50 m navd88 elevation in the 1870s were
converted using landfill into elevated neighborhoods and airports at 1.5-3.0 m NAVD88, and
thus are above this extra 0.20-0.28 m of storm tide. Similarly, for the United Kingdom the
frequency of extreme sea level events increased over the last 100 years, yet coastal flooding
hasn't increased (Haigh and Nicholls, 2017) because of improvements in forecasting/warning
and flood defenses.
It was previously established that the bay's tide ranges have grown substantially (Swanson and
Wilson, 2008), and we find similar results. Averaging high and low waters for daytime minima
and maxima in 1878 over 37 days gives an observed tide range of 1.35 m, while observations
for the entire year 2015 show a tide range of 1.73 m. This increase of 28% is smaller than the
prior estimate of the tide range change from 1899 to 2000 from Swanson and Wilson (2007),
which was 1.16 m to 1.64 m or 41%. However, the 1878 measurements are for a location at
mid-bay (Holland House), whereas the 1899 measurements are for the easternmost end of the
bay (Inwood or Norton Point), where tide attenuation (e.g. due to narrow, shallow channels
and wetlands) was likely more pronounced.


## 4. Discussion

In recent centuries, human activities have greatly modified the geomorphology and ecology of coastal regions, yet studies of historical and possible future changes in coastal flood extremes typically ignore the influence of geomorphic change (e.g., Lin et al., 2016; Orton et al., 2019). Jamaica Bay exemplifies an extreme case of "estuary urbanization" marked by land-fill, diking, channel deepening, and wetland loss (e.g., Marsooli et al., 2018). The upland changes reflected in **Figs. 4-5** and **Table 1** include widespread landfill and urbanization of fringe wetlands, the most visible result of these activities. Our results show that urbanization extends below the estuary water surface, with deepening of channels for shipping and excavation of borrow pits for landfill. The primary insight from this study that estuary urbanization amplifies storm tides likely applies to many urban sub-embayments worldwide, since basin engineering and wetland landfill for port development is globally a common and ongoing process (e.g., Murray et al., 2014; Paalvast and van der Velde, 2014; Schoukens, 2017).

Further analyses described below (**Sect. 4.1**) demonstrate that the specific changes to the bay that amplify storm tides (channel, inlet depths and widths, landfill) were all directly imposed by humans. Some contribution of the landscape and storm tide changes, such as the wetland erosion in the center of the bay, may be influenced by natural erosion or changing sediment supply (Peteet et al., 2018; Hu et al., 2018; Wang et al., 2017). However, the complex morphologic study required to separate these human and natural factors is beyond the scope of the present study. A broader discussion of the influence of the landscape changes on estuarine conditions and processes is given below (**Sect. 4.2**). Broader discussions of the multi-century landscape change at Jamaica Bay (**Sect. 4.3**) and the general implications of these results for dredged harbors and urbanized estuaries (**Sect. 4.4**) are also included herein.

### *4.1 Anthrogeomorphic amplification of storm tides*

The 1870s landscape mitigates storm tide elevations (**Fig. 6**) and damps them as they propagate into the bay (**Fig. 7**) by several potential mechanisms. First, the natural floodplain acts as a storage reservoir, allowing a given volume of water to spread over a larger area, but rising to a lesser vertical extent, than a confined (modern) system. Second, as also pointed out in Orton et al., (2015), the shallower historical channels and larger regions with marsh vegetation produced a more frictional environment that can damp long-waves such as tides and storm surge. Third, the shallower and narrower inlet may have altered the impedance of the storm surge into the estuary.


**Figure 7**: Maps of the 100-year flood for the present-day and 1870s landscapes. In both cases, floods were simulated with a 2016 mean sea level.




Simple "leverage experiments" are next used to isolate the effects of specific historical
landscape changes on the simulated water levels during a fast-moving, Category-3 hurricane
that approximates an event from 1821 (Orton et al., 2015). The storm surge from this hurricane
(3.4 +/- 0.4 m) likely exceeded the surge in hurricane Sandy (2.76 m), and produced water levels
of ~3m above 1821 mean sea-level despite occurring near low tide (Orton et al., 2016b).
Meteorological forcing for the simulations was created from parametric models (Orton et al.,
2015). The following experiments were performed using modifications to the modern-day
landscape to mimic the historical landscape's main features one-by-one:
• Tapered shallowing of the channel depth from offshore (8 m) into the inlet (5 m) and
10       into the innermost areas of the bay (1 m depth)
• Narrowing of the inlet so that its narrowest point is reduced by 50%
• Bay perimeter floodplain/wetland restoration, including reducing elevation and altering
friction coefficients to represent wetland land cover
• Wetland restoration in the center of bay to the 1870s footprint
• Inclusion of additional roughness, to mimic effect of eelgrass and oyster shells
• Restoration of a shoal off the west end of Rockaway Peninsula
• Shallowing the deep borrow pit area on the northeast side of the bay
• Restoration of the landform to the north of the inlet to wetlands
• Narrowing channels on the interior of the bay
Three of the leverage experiments led to large reductions in hurricane storm tide. The tapered
shallowing leads to a change in the peak hurricane storm tide of -56 cm or -23% (**Fig. 8ab**). The
inlet narrowing leads to a change of -19 cm or -8% (**Fig. 8cd**). Bay perimeter floodplain/wetland
restoration results in a change of -13% (**Fig. 8ef**). All the other landscape changes showed
smaller impacts, indicating that they likely play little role in the long-term changes to storm
tides. For example, extensive wetland restoration in the center of the bay only leads to a
change in peak storm tide of only -2%, because deep shipping channels around the wetlands
are the primary conduit for flood waters (Orton et al., 2015). A small rise in Mannings-n to
0.025 (mimicking scattered areas of lost eelgrass or shells) reduced the peak by -3%. The other
changes also had relatively minor effects.
Scaling suggests that the conveyance of long waves (e.g. storm surges, tides) through an inlet
into a lagoonal estuary depends on the inlet choking number $P = \left(\frac{gb^2H^3T^2}{C_d L \eta A_e^2}\right)^{1/2}$, i.e., on the drag
coefficient ($C_d$), inlet width (b), length (L), depth (H), tide or surge amplitude ($\eta$), the long-wave
period (T), and estuary surface area ($A_e$) (e.g., MacMahan et al., 2014; Stigebrandt, 1980). For
decreasing value of P, the inlet is increasingly "choked", meaning that long-wave amplitudes
strongly decrease entering the lagoon. For low values of P (below 5), choking becomes
important, and for high P (above 10), the inlet geometry is unimportant (Stigebrandt, 1980).
The dependence of P on $H^{3/2}$ conveys a strong sensitivity to water depth, and dependencies on
b and $A_e$ convey modest sensitivities to inlet width and estuary area.

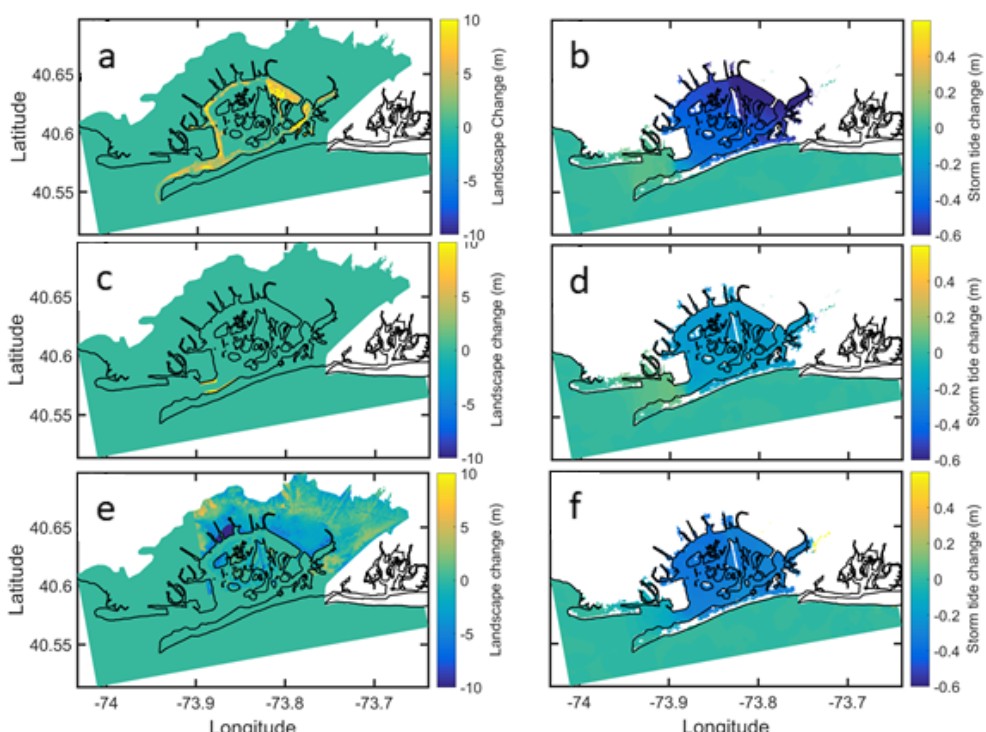

**Figure 8**:  Results of "leverage experiments" used to isolate the effects of specific historical
landscape changes, testing their influence on the storm tide for a Category-3 hurricane. The left
panels (a,c,e) show the imposed changes made to the present-day landscape, where the black
line shows the present-day coastline. The right panels (b,d,f) show the resulting modeled storm
tide changes. The top row is channel shallowing, middle row inlet narrowing, and bottom row
interior floodplain restoration.
Our landscape reconstruction and numerical results suggest that the choking of long waves at
Rockaway Inlet has been strongly reduced. For typical tides, we estimate that P increased from
4.5 in the 1870s to 13 at present. For a large amplitude, short-timescale storm surge such as the
1821 hurricane, P has changed from 0.69 to 2.0. These changes are driven by a 41% increase in
inlet's average depth (from 6.0 to 8.5 m), and 50% increase in average width (from 1000 to
1500 m), and a 23% reduction in bay area. A lengthening of the inlet (from 6600 to 9900 m) due
to the growth of Rockaway Peninsula slightly counteracts these effects on choking number,
however. Measured at its minimum along-inlet location, there is an 85% increase in the cross-
sectional area of the inlet, from 4800 to 8900 m$^2$. Reflection and possibly resonance likely play a
role in the amplification of tides in the present-day estuary, whereas the shallow water depths
and frictional effects of fringing wetlands would also reduce these effects in the 1870s system.



The dependence of the inlet choking number on both geometric properties and long-wave
characteristics helps interpret numerical results. Changes to inlet geometry and channel depth
have most strongly changed the large impact, high amplitude storm surges caused by TCs such
as the 1821 event. Smaller amplitude events caused (e.g. ETCs) are less likely to be affected by
inlet geometry; this is one of the reasons that there is a lesser change in the 5-year storm tide
than the 100-year storm tide (**Fig. 6**). The difference between the 500-year storm tide for 1877
and present-day landscapes is not larger than that or the 100-year storm tide. This may arise
because overtopping of Rockaway Peninsula becomes important, circumventing the inlet and
invalidating the above scaling arguments.
Similarly, the tide or surge time-scale (wave period) T impacts the conveyance of surge or tide
into estuaries and back-bays (Aretxabaleta et al., 2017; Kennedy et al., 2011), and the damping
that occurs within them (Orton et al., 2015). Slow surge events such as Hurricane Sandy (e.g.,
those building to a peak over more than 18 hours) are less affected by hydrodynamic drag (due
to smaller flow velocity), potentially producing more severe estuarine floods (Familkhalili and
Talke, 2016; Orton et al., 2015). These considerations suggest that modeling flood hazard or
designing infrastructure using a representative "storm of record" can produce bias; instead,
using an ensemble approach (such as used here) with both small and large time-scale events
produces better results.
The primary reasons for increased storm tides – the floodplain (bay area) reduction, inlet width
and depth increases, and bay channel depth increases – were all imposed by human activities
such as landfilling, dredging, inlet stabilization (e.g. with the jetties) and shoreline hardening.
Therefore, we conclude that the amplification in storm tides is primarily of anthropogenic
origin.
***4.2 Ecological importance of landscape changes since the 1870s***
The present-day landscape of Jamaica Bay supports a highly eutrophic, but in many ways
healthy estuarine ecosystem with oxygen levels slowly rising over recent decades (Walsh et al.,
2018; NYC-DEP, 2018). However, various indicator species, particularly those that depend on
intertidal habitats (e.g. diamondback terrapin) have continued declining in abundance (Walsh
et al., 2018). Here, we note some likely ecological influences of the landscape and habitat
changes summarized in **Figs. 4-5** and **Table 1**.
Our landscape reconstruction confirms that the bay's eelgrass beds have disappeared
completely, and wetland area has declined dramatically since the 1800s. The wetland decline
may be stopped with marsh island restoration/ reconstruction activities which have been
occurring over the past decade (Seavitt et al., 2015). Eelgrass beds provide many similar
ecological services in estuaries, including nursery and refuge for a diverse and dense faunal
community, trapping of sediment, and erosion prevention (Orth et al., 2006). They are known
to decline in eutrophic conditions due to reduced sunlight (Vaudrey et al., 2010; Orth et al.,
2006). Salt marshes are widely-known for their ecological importance, including many of the
same roles as eelgrass beds, but including intertidal habitat. At Jamaica Bay, this habitat serves
diamondback terrapin and birds such as the sharp-tailed sparrow, egrets, herons and geese.





Our landscape reconstruction shows that unvegetated intertidal area has decreased by 12.7 km$^2$ , a loss of 74% . This change is of equal magnitude in km$^2$ to the loss of eelgrass beds (**Table 1**). Mudflats, sandbars, oyster and mussel reefs, and other unvegetated intertidal areas are forms of "shallows", and provide important habitat for benthic invertebrates like polychaetes, snails, clams, crabs, and blue mussels, as well as birds that feed on them such as the oystercatcher and willet. They are also used by terrapins for feeding and by horseshoe crabs for reproduction.

The center of the bay (inside the channels that circle the bay today) has not only lost marsh islands, it has had its land elevation drop substantially, most areas by about 1 m since the 1800s (**Fig. 5**). What were once large expanses of intertidal unvegetated area have shifted to being subtidal. This drop may reduce the sediment supply to the remaining marsh islands' substrate during storms (Wang et al., 2017). Also, an increased depth in front of the marsh can increase wave energy and promote lateral erosion (Fagherazzi et al., 2006). As a result, the loss of intertidal zones and associated increased water depths may be detrimental to the sustainability of the remaining marsh islands and their critical habitat.

The increase from 7 to 28 km$^2$ of deep habitat areas (**Table 1**) may attract more large fish such as striped bass due to increased swimming space, the reduction in thermal variability caused by a deep water column, or stratified deep water's lower temperature in summertime. It is unknown whether there were more or less striped bass in Jamaica Bay in the 1800s, but their presence today has the benefit of supporting a small fleet of fishing charter boats. However, there are several square kilometers of poorly-flushed deepwater regions, predominantly in Grassy Bay immediately southwest of Kennedy Airport, that are prone to hypoxia and even anoxia in late summer, providing compromised habitat area for many organisms (NYC-DEP, 2018).

Our landscape reconstruction and modeling suggest that the residence time of water within the bay has more than doubled between the 1870s and today, with potential adverse ecological implications. The residence time of water in an estuary that receives large wastewater-derived nutrient inputs like Jamaica Bay is an important control on hypoxia, with longer residence times often leading to worsened hypoxia (e.g., Sanford et al., 1992). A simple model of the residence time of a lagoonal-type estuary system is the volume of the bay divided by the tidal flux rate which is the tide prism (volume of water between mean high water and mean low water) over the tide period (12.42 hours) (e.g., Sanford et al., 1992). For the 1870s landscape and sea level, the average modeled tidal prism of the bay was 80 million m$^3$ and volume was 97 million km$^2$, leading to a residence time of 0.62 days. For the 2015 landscape and sea level, the average modeled tidal prism of the bay was 102 million m$^3$ and volume was 290 million m$^3$, leading to a computed residence time of 1.5 days. This simple model was shown for the modern landscape to underestimate residence times (relative to modeled tracer releases), but nevertheless shows that the changes in bay morphology lead to a substantial increase in residence time of a factor of 2-3 mainly due to the much greater volume of the present-day bay. More detailed analyses of water quality and residence time have been performed in other recent studies, and these




results are being reported on in separate papers but generally support this interpretation
(Marsooli et al., 2018; Fischbach et al., 2018).
***4.3 Earlier Jamaica Bay landscapes: The estimated 1609 landscape***
The 1870s landscape of Jamaica Bay was already influenced by humans. Prior to European
colonization, Jamaica Bay was likely more open to the ocean, with an actively migrating inlet
located further to the east, a barrier island system, extensive fringing marshlands, but far fewer
marsh islands than in the 1870s (Black, 1981; Sanderson, 2016). A less well-constrained model
for the pre-European landscape was also produced for this study, and modeling suggests storm
tide reductions (from offshore into the bay) were caused by the landscape of the 17th century
(Orton et al., 2016a). The model was based on 17th and 18th century maps that showed
coastlines and major features, such as an inlet which was in the center of today's Rockaway
Peninsula, and a general absence of marsh islands in the bay, calling the bay "Jamaica Sound".
However, the maps did not show bathymetry measurements, and therefore the actual
hydrodynamic behavior of the system is highly uncertain relative to the 1870s and present-day
landscape (Orton et al., 2016a). Ongoing research is helping improve our understanding of the
landscape of the 1600s and long-term evolution through analyses of sediment cores from the
western-central area and eastern ends of the bay (Peteet et al., 2018). That study showed that
European settlement led to increases of inorganic sediment delivered to the bay, likely due to
forest clearance for agriculture and subsequent erosion, and this may explain the increase in
marsh island area in the 1700s and 1800s. These considerations suggest that on century
timescales, hard to quantify factors such as the anthropogenically mediated sediment supply
may also exert an important influence on long-term system evolution.
***4.4 Broader context***
Remarkably, despite the visions of the Jamaica Bay Improvement Commission (1907) and The
River and Harbor Acts of 1910 and 1925, the present-day commercial shipping activity through
this largely man-made 1 km wide, 8-16 m deep shipping channel (measured at Floyd Bennett
Field, the narrowest part) is limited to an average of 3 one-way trips per day servicing
gravel/sand companies, sewage treatment plants and bulk fuel companies (USACE, 2016). Our
results show that maintaining these shipping channels leads to higher storm tides in the bay,
even though the economic activity that justified their construction is largely absent.
Globally, common development approaches such as dredging for port development and
landfilling for neighborhood development can have major economic benefits, but can also raise
vulnerability as they did for Jamaica Bay (Talke and Jay, 2020). The movement towards "New-
Panamax" and larger ships is leading major harbors to dredge wider channels and depths of
approximately 16 m (Briggs et al., 2015). Other dredged estuaries have been shown to cause
enhanced inland propagation of storm tides, such as with the Cape Fear estuary (Familkhalili et
al. 2016). The Mississippi Gulf River Outlet canal was originally created through dredging, and
recently was de-authorized and blocked in part because of a debate over whether it increased
storm surge penetration inland (Shaffer et al., 2009). Within the St Johns River, Florida, channel
deepening to a controlling depth of >14m is continuing, despite model results that showed
increases in tide range and storm surge of 0.1-0.2 m in some locations (USACE, 2014).



The results presented here suggest that evaluating changes to flood hazard should be part of
the cost-benefit analysis of any environmental impact study or restoration study, particularly
projects that propose altering inlet geometry or channel depth. Our results can help inform
debates about whether to continue maintaining under-used ports, since allowing inlets and
channel depths to return to pre-development geometry is potentially a way to mitigate against
future sea-level rise effects. Given adequate sediment supply, many systems quickly return
towards pre-development depths; for example, the lower Passaic River in New Jersey has
accumulated as much of 5 m of sediment after maintenance dredging ceased in the early 1980s
(Chant et al., 2011).
**5. Conclusions**
This study applied a historical reconstruction approach for a case study of how natural and
urbanized estuary systems modify coastal storm tides. A Jamaica Bay flood model for the 1870s
was developed and simulation results were contrasted with those from a present-day model to
quantify the influences of 20th Century changes in bathymetry and habitat on storm tide hazard.
The hydrodynamic model landscape (land elevation and friction) for the 1870s was estimated
from detailed maps of topography, bathymetry and seabed characteristics, and validated using
tide observations. The models were used for tide simulations, supplementing map data with
tidal datums for additional analysis of habitat change (e.g. estuary intertidal area), and for
coastal storm flood modeling and probabilistic hazard assessment.
Major changes to land elevation and land cover were quantified and translated into habitat
area changes, more precisely constraining previous estimates of mean depth change and
previously-reported estimates of marsh loss. Predominantly through dredging, landfill and inlet
stabilization, the average water depth of the Jamaica Bay has increased from 1.7 to 4.5 m, tidal
surface area diminished from 92 to 72 km$^2$, and the inlet cross-sectional area was expanded
from 4800 to 8900 m$^2$. Total (freshwater plus salt) marsh habitat area was estimated to decline
by 74%, intertidal area by 73%, and intertidal unvegetated habitat area by 72%, both by about a
factor of four. Deepwater habitat increased by 314%, also about a factor of four. Submerged
grasses (e.g. eelgrass) disappeared completely.
A probabilistic flood hazard assessment with simulations of 144 storm events revealed that the
landscape changes caused an increase of 0.28 m (12%) in the 100-year storm tide, similar to the
separate effect of a global sea level rise of 0.23 m (Church and White, 2011; Hay et al., 2015)
and local sea level rise of 0.37 m from the 1870s to 2015 (Kemp and Horton, 2013). The 10-year
storm tide increased by 0.20 m (11%). In spite of these rising storm tides, flood area for the 10-
year and 100-year storm tides is smaller than it was in the 1870s, by 19 and 14%, respectively,
due to landfill conversion of fringing wetlands into elevated neighborhoods.
Specific anthropogenic changes to estuary depth, area and inlet depth and width were shown
through targeted modeling and dynamics-based considerations to be important drivers of these





changing storm tides, with depth changes being the strongest factor. The dependence of inlet
choking of a long-wave such as tide or surge depends on estuary area squared, inversely on
inlet width squared, and inversely on inlet/estuary depth cubed. These choking effects are also
enhanced with short-duration sea level anomalies, such that a rapid-pulse storm surge rising in
a matter of a few hours is damped more than a semidiurnal tide or long-duration storm surge
event.  Similar scaling shows that damping within the estuary has also decreased.
Our study highlights that anthropogenic changes to estuary geomorphology can affect storm
tide hazard to a degree that is comparable to historical sea-level rise. An improved
understanding of historical estuarine landscapes, as well as their hydrodynamic and
sedimentary processes, can help inform nature-based flood and climate mitigation efforts.
Studies such as this one that reconstruct the historical landscape can be used to assess
strategies to minimize floods into the future, as demonstrated on the broader nature-based
adaptation study (Orton et al. 2016) website and flood adaptation mapper tool
(http://AdaptMap.info). These results have influenced adaptation considerations after
Hurricane Sandy spurred a strong interest in flood adaptation. Concepts of bay shallowing and
inlet narrowing were considered as options in a stakeholder-driven study of nature-based
options for flood and hypoxia mitigation, with narrowing being one of the more deeply-
evaluated alternatives (Fischbach et al., 2018).
*Data availability.*  Model DEMs, still-water elevation data, and animations of model simulations
for the 1870s and present-day are available by download from the project's flood mapper
http://AdaptMap.info/jamaicabay/ (5-year through 1000-year still-water elevation, in GeoTIFF
and CSV formats). Observed tide data for 1877-1878 are available at the US National Archives in
College Park, MD, in Record Group 23, Entry 148, PI. 105.  Tide data used for 2015 are available
from the United States Geological Survey (station 01311850) via https://waterdata.usgs.gov.
*Author contributions.*  Conceptualization, PMO and EWS; Methodology, PMO; Formal Analysis,
PMO; Data Curation, PMO; Writing-Original Draft Preparation, PMO; Writing-Review & Editing,
PMO, EWS and SAT; Visualization, PMO, EWS and MG; Project Administration, PMO; Funding
Acquisition, PMO, EWS, KM and SAT.
*Competing interests.*  The authors declare that they have no conflict of interest.
*Financial support.*  PMO was funded by the United States (US) National Science Foundation
(NSF PREEVENTS award 1855037) and National Oceanic and Atmospheric Administration (NOAA
NA16OAR4310157). EWS, MG and KM were funded by NOAA (NA13OAR4310144). SAT was
funded by NSF (Career Award 1455350), the US Army Corps of Engineers (Award W1927N-14-2-
0015) and NSF PREEVENTS (1854946).



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

USACE: Final Integrated General 708 Reevaluation Report II and supplemental Environmental
Impact Statement. Jacksonville Harbor, Duval County, Florida. United States Army Corps of
Engineers, Jacksonville District. Appendix A, Attachment J, p.1623-1646., 2014.
USACE: Atlantic Coast of New York, East Rockaway Inlet to Rockaway Inlet and Jamaica Bay:
Draft Integrated Hurricane Sandy General Reevaluation Report and Environmental Impact
Statement, United States Army Corps of Engineers, New York District, New York, NY, 270 pp.,
22  2016.

Vaudrey, J. M., Kremer, J. N., Branco, B. F., and Short, F. T.: Eelgrass recovery after nutrient
enrichment reversal, Aquatic Botany, 93, 237-243, 2010.
Wahl, T., and Chambers, D. P.: Climate controls multidecadal variability in US extreme sea level
records, J. Geophys. Res., 2016.
Walsh, B., Costanzo, S., and Taillie, D.: Natural Resource Condition Assessment, Gateway
National Recreation Area, Natural Resource Report NPS/GATE/NRR—2018/1774, Department
of the Interior, National Parks Service, 170 pp., 2018.
Wang, H., Chen, Q., Hu, K., Snedden, G. A., Hartig, E. K., Couvillion, B. R., Johnson, C. L., and
Orton, P. M.: Numerical modeling of the effects of Hurricane Sandy and potential future
hurricanes on spatial patterns of salt marsh morphology in Jamaica Bay, New York City, US
Geological Survey2331-1258, 2017.
Warner, J. C., Geyer, W. R., and Lerczak, J. A.: Numerical modeling of an estuary: A
comprehensive skill assessment, J. Geophys. Res., 110, 10.1029/2004jc002691, 2005.
Wilson, H. M.: Hempstead NY Quadrangle, U.S. Geological Survey, Washington, D.C., 1897.