# Peer review of "Storm Tide Amplification and Habitat Changes due to Urbanization of a Lagoonal Estuary"

_Natural Hazards and Earth System Sciences, 2019_

## Referee Comment (RC1) · Anonymous Referee #1 · 3 Mar 2020

This paper compiles historic datasets on land cover, topography and coastal water levels and combines these with statistical and coastal models of extreme water levels and storm tides to understand how human modifications have changed the signal of extreme water levels in Jamaica Bay between 1870 and now. The paper is well-written and addresses a highly relevant topic. My main comments on the paper are:

1. the authors should include at least 1 paragraph outlining the underlying processes incorporated by the seCOM model. While this information exists in the cited publications (e.g. Orton et al., 2016), it will be helpful to have some of this information in this paper as well.

2. In Section 2.3, the authors should include a few sentences describing how the synthetic storm set from Orton et al., 2016b was abbreviated for the purposes of this

study. How were the random tide permutations for the ETCs or the storm tide events for the TCs abbreviated?

3. I recommend moving the results from the leverage experiments described in Section 4.3. to Section 3, Results.

4. Sentence on Page 3, L13 is incomplete.

---

## Referee Comment (RC2) · Carmine Donatelli (Referee) · 20 Apr 2020

I have read with great interest the paper entitled: 'Storm Tide Amplification and Habitat Changes due to Urbanization of a Lagoonal Estuary', by Orton et al.

The manuscript is very well written and organized, the used methodology is robust and supported by the clearness of the presented data. Previous works are referenced appropriately.

I think this study represents an important contribution to our understanding of the mechanisms by which geomorphic changes alter the physical response of back-barrier estuaries to tides and low frequency actions. I recommend publication with minor revisions.

My minor comments/suggestions are listed below.

[Figure]

Pages 10-11: 'The most dramatic land cover change is from large areas of fringing wetlands (light blue) to urbanized areas (red), but also the center of the bay has shifted from marshes to open waters (1 dark blue)'.

The influence of marsh loss on water levels is strongly related to the setting: marsh loss associated with lateral erosion vs. marsh loss associated with reclamation projects. Reductions in the basin planform area (wetland reclamation) increase water levels, while marsh retreat due to lateral erosion may have the opposite effect (in agreement with the analytical model of Keulegan, 1967 and 3D numerical modelling investigations, Picado et al., 2010; Donatelli et al., 2018). I would highlight/expand this point in the text.

Page 15: 'It was previously established that the bay's tide ranges have grown substantially (Swanson and Wilson, 2008), and we find similar results. Averaging high and low waters for daytime minima and maxima in 1878 over 37 days gives an observed tide range of 1.35 m, while observations for the entire year 2015 show a tide range of 1.73 m. This increase of 28% is smaller than the prior estimate of the tide range change from 1899 to 2000 from Swanson and Wilson (2007), which was 1.16 m to 1.64 m or 41%. However, the 1878 measurements are for a location at mid-bay (Holland House), whereas the 1899 measurements are for the easternmost end of the bay (Inwood or Norton Point), where tide attenuation (e.g. due to narrow, shallow channels and wetlands) was likely more pronounced.'

I would move these lines into the Discussion section.

Page 16 (lines 11-14): I would expand these lines. Are the results for Jamaica Bay translatable to other systems? What systems?

Page 18 (lines 1-19): I would move the description of the idealized numerical experiments into the Methods section.

Page 21: I would mention Englebright [1975] and Harting et al., [2002], who show how

inlet modifications have reduced the movement of offshore sediments into the back-barrier basin.

---

## Referee Comment (RC3) · Anonymous Referee #3 · 27 Apr 2020

General comments: The paper entitled "Storm Tide Amplification and Habitat Changes due to Urbanization of a Lagoonal Estuary" by Philip M. Orton et al. analyzes the impacts of historic landscape changes within the 20th century in Jamaica Bay, New York City on present-day storm tide water levels. The results of this study reveal considerable effects of especially channel deepening on storm tide peak water levels within the bay, which is comparable to, and even exceeding, global mean sea level rise during the same period. The work carried out by Orton et al. is scientifically solid, well written and the presented results are highly relevant not only to Jamaica Bay, New York City, but also to many other highly populated and developed coastal areas around the globe. The methods are generally sound (some issues are listed below) and the conclusions are supported by the presented data. I recommend publication of this article with minor

revisions.

My detailed comments are listed below:

Page 7, lines 15 – 19: I think this part needs some clarification. I would doubt that the landward edge of a saltmarsh equals the extent of high tide flooding. This would mean that the marsh gets completely flooded during most of the tides, which should not be the case. The upper edge of the high salt marsh is usually only flooded during storm tides or the highest spring tides. What are the implications of this approach to the validity of the results? Something, which could also be taken up in the discussion.

Page 7, line 29: How was the digitized bathymetric and topographic data resampled in order to meet the named 30 m grid resolution (line 2 page 6)? Which resampling technique was used (e.g. bilinear, nearest or cubic etc.)?

Page 8, lines 7 – 9: Briefly outlining the methodology used to simulate the ensemble of storm tides would help the reader. You could still write that a detailed description is given in Orton et al. 2016b.

Page 8, lines 9-12: I would appreciate some more information on the functioning and structure of the model sECOM. What exactly does accurate mean? Can you provide an estimate of the error associated with it?

Page 8, lines 39-42: Does this mean you have assigned a single Manning n coefficient to both eelgrass and saltmarsh vegetation? In the literature there are many studies suggesting Manning n coefficients of 0.07 (Lawrence et al. 2004) or up to 0.08 for salt-marsh surfaces (Stark et al. 2016, 2017; Temmerman et al. 2012) and I assume that the roughness of seagrass beds should be considerably less. What was the decision to take a value of 0.045 based on? What is the sensitivity of your model to variations in these coefficients? I think answering these questions is important, since you state that the "most dramatic land cover change is from large areas of fringing wetlands (light blue) to urbanized areas (red)" (page 10, lines 43 – 44) and because artificially

recovering the wetlands in your model has only resulted in a reduction of peak storm surge heights of -2% (page 18 lines 26-28). On the other hand, increasing Manning n coefficients to 0.025 for scattered areas of lost eelgrass resulted in a peak reduction of 3%. This needs some further explanations.

Page 14, lines 17-20: Please consider moving this sentence to the discussion section, as you start interpreting your results here.

Page 15, lines 25 – 33: This part should be moved into the discussion too.

Page 20, lines 40-41: I suggest rephrasing this sentence to make it a little clearer: It is not the sunlight that is reduced but due to increased turbidity, the light penetration into the water column is reduced.

Figure 4: Perhaps it is just due to the system that created the pdf file, but the legends of both maps are very hard to read. Please check and increase the size of the key.

Literature

Lawrence, D.S.L.; Allen, J.R.L.; Havelock, G. M. (2004): Salt Marsh Morphodynamics: an Investigation of Tidal Flows and Marsh Channel Equilibrium. In: Journal of Coastal Research 20 (1); p. 301-316

Stark, Jeroen; Plancke, Yves; Ides, Stefaan; Meire, Patrick; Temmerman, Stijn (2016): Coastal flood protection by a combined nature-based and engineering approach. Modeling the effects of marsh geometry and surrounding dikes. In: Estuarine, Coastal and Shelf Science 175; p. 34-54.

Stark, J.; Smolders, S.; Meire, P.; Temmerman, S. (2017): Impact of intertidal area characteristics on estuarine tidal hydrodynamics. A modelling study for the Scheldt Estuary. In: Estuarine, Coastal and Shelf Science 198; p. 138-155.

Temmerman, Stijn; Vries, Mindert B. de; Bouma, Tjeerd J. (2012): Coastal marsh die-off and reduced attenuation of coastal floods. A model analysis. Global and Planetary

[Figure]

Change 92-93; p. 267-274.

---

## Author Comment (AC1) · 9 Jun 2020

The authors thank the reviewers for a very detailed reading of the paper and substantive comments that have clearly improved the research and its presentation. Below are the original comments and our responses below them. The responses are also uploaded as a supplement PDF with better differentiation.

Anonymous Referee #1 This paper compiles historic datasets on land cover, topography and coastal water levels and combines these with statistical and coastal models of extreme water levels and storm tides to understand how human modifications have changed the signal of extreme water levels in Jamaica Bay between 1870 and now. The paper is well-written and addresses a

highly relevant topic.

My main comments on the paper are: 1. the authors should include at least 1 paragraph outlining the underlying processes incorporated by the seCOM model. While this information exists in the cited publications (e.g. Orton et al., 2016), it will be helpful to have some of this information in this paper as well.

Change made – We have added a paragraph that elaborates on the model: "The Stevens Estuarine and Coastal Ocean Model (sECOM) is a free-surface, hydrostatic, primitive equation model, with terrain-following (sigma) vertical coordinates, set on an orthogonal, curvilinear Arakawa C-grid (Georgas and Blumberg, 2010; Blumberg et al., 1999). The model has been further developed with regard to wind stress formulations (Orton et al., 2012), coupled wave modeling (Georgas et al., 2007), and land wetting and drying (Blumberg et al., 2015). It has been used to provide validated and accurate ensemble 3D storm tide predictions as part of the NY Harbor Observation and Prediction System (NYHOPS; Georgas and Blumberg, 2010) and the Stevens Flood Advisory System (Jordi et al., 2018). Typical errors in hindcasts of extreme storm tides (e.g. Hurricane Sandy) are 0.15-0.20 m (Orton et al. 2016)."

2. In Section 2.3, the authors should include a few sentences describing how the synthetic storm set from Orton et al., 2016b was abbreviated for the purposes of this study. How were the random tide permutations for the ETCs or the storm tide events for the TCs abbreviated?

Changes were made to give more detail, revising the text to read: "The abbreviated set of 80 ETCs includes all the same storm events, but fewer random tide permutations for each storm. Instead of 50 simulations for the top 19 historical ETC storm tide events, there were 5 or 10 simulations each for the 11 highest ETC storm tides that are most relevant for the 5-year and higher return periods. The abbreviated set of 64 TCs includes a range of storm tide events from low to high magnitude (1.5 to 6.0 m). Model results for simulated TC events at a given magnitude are then used as a proxy for all

the events at that magnitude, thus representing all 606 storms."

3. I recommend moving the results from the leverage experiments described in Section 4.3. to Section 3, Results.

Change made – I kept hearing this and capitulated – see Section 3.3. The methods were also moved to the methods section (Section 2.4).

4. Sentence on Page 3, L13 is incomplete.

Change made, fixing the error.

Please also note the supplement to this comment:
https://www.nat-hazards-earth-syst-sci-discuss.net/nhess-2019-343/nhess-2019-343-AC1-supplement.pdf

---

## Author Comment (AC2) · 9 Jun 2020

The authors thank the reviewers for a very detailed reading of the paper and substantive comments that have clearly improved the research and its presentation. Below are the original comments followed by our responses below them, and a supplemental PDF gives them with better differentiation (italics vs plain text).

Carmine Donatelli (Referee) c.donatelli@liverpool.ac.uk I have read with great interest the paper entitled: 'Storm Tide Amplification and Habitat Changes due to Urbanization of a Lagoonal Estuary', by Orton et al.

The manuscript is very well written and organized, the used methodology is robust and supported by the clearness of the presented data. Previous works are referenced

appropriately.

I think this study represents an important contribution to our understanding of the mechanisms by which geomorphic changes alter the physical response of back-barrier estuaries to tides and low frequency actions. I recommend publication with minor revisions.

My minor comments/suggestions are listed below.

Pages 10-11: 'The most dramatic land cover change is from large areas of fringing wetlands (light blue) to urbanized areas (red), but also the center of the bay has shifted from marshes to open waters (1 dark blue)'.

The influence of marsh loss on water levels is strongly related to the setting: marsh loss associated with lateral erosion vs. marsh loss associated with reclamation projects. Reductions in the basin planform area (wetland reclamation) increase water levels, while marsh retreat due to lateral erosion may have the opposite effect (in agreement with the analytical model of Keulegan, 1967 and 3D numerical modelling investigations, Picado et al., 2010; Donatelli et al., 2018). I would highlight/expand this point in the text.

Text was added to address this useful point – These results are also consistent with prior studies that showed that the influence of lagoonal wetland loss on water levels is different when it comes to lateral erosion versus landfill reclamation. Reductions in the tidally-wetted area through wetland reclamation increase storm tides, while wetland retreat due to lateral erosion has the opposite effect (e.g., Donatelli et al., 2018; Picado et al., 2010).

Page 15: 'It was previously established that the bay's tide ranges have grown substantially (Swanson and Wilson, 2008), and we find similar results. Averaging high and low waters for daytime minima and maxima in 1878 over 37 days gives an observed tide range of 1.35 m, while observations for the entire year 2015 show a tide range of 1.73 m. This increase of 28% is smaller than the prior estimate of the tide range

change from 1899 to 2000 from Swanson and Wilson (2007), which was 1.16 m to 1.64 m or 41%. However, the 1878 measurements are for a location at mid-bay (Holland House), whereas the 1899 measurements are for the easternmost end of the bay (Inwood or Norton Point), where tide attenuation (e.g. due to narrow, shallow channels and wetlands) was likely more pronounced.'

I would move these lines into the Discussion section.

This paragraph actually gives results – it is the first time tide range changes are presented. Moreover, the discussion and primary paper focus is on the topic of storm tides. Thus, we have not made this change.

Page 16 (lines 11-14): I would expand these lines. Are the results for Jamaica Bay translatable to other systems? What systems?

Text was added to address this suggestion: "Systems with likely impacts include those with substantial changes to inlets, mean estuary depths, or wetland landfill/reclamation, and may be detected by long-term changes to tides (Talke and Jay, 2020)."

Page 18 (lines 1-19): I would move the description of the idealized numerical experiments into the Methods section.

Change made – I kept hearing this and capitulated – see Section 2.4. The results were also moved to the results section (Section 3.3).

Page 21: I would mention Englebright [1975] and Harting et al., [2002], who show how inlet modifications have reduced the movement of offshore sediments into the backbarrier basin.

This suggested change was not made, as these studies were speculative about changes to the inlet's role in sediment import to the bay. I'd rather not further the speculation, because there is a study of sediment fluxes into the bay that I hear is in review.

Please also note the supplement to this comment:
https://www.nat-hazards-earth-syst-sci-discuss.net/nhess-2019-343/nhess-2019-343-AC2-supplement.pdf

---

## Author Comment (AC3)

The authors thank the reviewers for a very detailed reading of the paper and substantive comments that have clearly improved the research and its presentation. Below are the original comments in italics and our responses below them.

**Anonymous Referee #3**

*General comments: The paper entitled "Storm Tide Amplification and Habitat Changes due to Urbanization of a Lagoonal Estuary" by Philip M. Orton et al. analyzes the impacts of historic landscape changes within the 20th century in Jamaica Bay, New York City on present-day storm tide water levels. The results of this study reveal considerable effects of especially channel deepening on storm tide peak water levels within the bay, which is comparable to, and even exceeding, global mean sea level rise during the same period. The work carried out by Orton et al. is scientifically solid, well written and the presented results are highly relevant not only to Jamaica Bay, New York City, but also to many other highly populated and developed coastal areas around the globe. The methods are generally sound (some issues are listed below) and the conclusions are supported by the presented data. I recommend publication of this article with minor revisions.*

*My detailed comments are listed below:*
*Page 7, lines 15 – 19: I think this part needs some clarification. I would doubt that the landward edge of a saltmarsh equals the extent of high tide flooding. This would mean that the marsh gets completely flooded during most of the tides, which should not be the case. The upper edge of the high salt marsh is usually only flooded during storm tides or the highest spring tides. What are the implications of this approach to the validity of the results? Something, which could also be taken up in the discussion.*

The text was unclear, and changes have been made to clarify. We actually did use the annual highest astronomical tide (HAT) as the elevation of the upper edge of the high salt marsh. HAT was estimated using 1840s tide data from a nearby location with a similar tide range (Governor's Island, New York Harbor). Text has been revised to change "high tide flooding" to read "highest astronomical tide flooding".

*Page 7, line 29: How was the digitized bathymetric and topographic data resampled in order to meet the named 30 m grid resolution (line 2 page 6)? Which resampling technique was used (e.g. bilinear, nearest or cubic etc.)?*

Change made to clarify – Bilinear interpolation was used, and this word is added to the text at that location.

*Page 8, lines 7 – 9: Briefly outlining the methodology used to simulate the ensemble of storm tides would help the reader. You could still write that a detailed description is*

*given in Orton et al. 2016b.*

Change made - the detailed description is actually given in the subsequent section, and the text now refers to Section 2.3 for that information: "A hydrodynamic model was applied to the historical and modern "landscapes" (land surface elevation and roughness) and used to simulate an ensemble of storm tide events described in Section 2.3."

*Page 8, lines 9-12: I would appreciate some more information on the functioning and structure of the model sECOM. What exactly does accurate mean? Can you provide an estimate of the error associated with it?*

Changes made –
"The Stevens Estuarine and Coastal Ocean Model (sECOM) is a free-surface, hydrostatic, primitive equation model, with terrain-following (sigma) vertical coordinates, set on an orthogonal, curvilinear Arakawa C-grid (Blumberg et al. 1999; Georgas and Blumberg, 2010; Orton et al. 2012) … Typical errors in hindcasts of extreme storm tides (e.g. Hurricane Sandy) are 0.15-0.20 m (Orton et al. 2016)."

*Page 8, lines 39-42: Does this mean you have assigned a single Manning n coefficient to both eelgrass and saltmarsh vegetation? In the literature there are many studies suggesting Manning n coefficients of 0.07 (Lawrence et al. 2004) or up to 0.08 for saltmarsh surfaces (Stark et al. 2016, 2017; Temmerman et al. 2012) and I assume that the roughness of seagrass beds should be considerably less. What was the decision to take a value of 0.045 based on?  What is the sensitivity of your model to variations in these coefficients?*

A model simulation was performed to quantify the uncertainty.  We were unable to find a Manning's-n number in the literature for eelgrass (Zostera Marina).  As eelgrass and other macrophytes are widely accepted to reduce flow rates, we chose a simple route and set the Mannings number to be the same as Spartina Alterniflora.  In retrospect, I agree that a lower value could have been better.  To address the resulting possible bias, I ran an experiment to quantify the effect on one sample 100-year storm tide event, a hurricane, and found the maximum difference occurred in the northern bay and was a reduction of 8 millimeters for peak storm tide.  This is because the eelgrass area is only 32% of the bay interior benthic substrate (below MSL elevation), and the flow during a 100-year storm tide event is 3-4m deep over this roughened surface.  I conclude that the choice of a high eelgrass mannings-n value does not significantly alter our results, nor the main conclusions of the paper.

*I think answering these questions is important, since you state that the "most dramatic land cover change is from large areas of fringing wetlands (light blue) to urbanized areas (red)"*

*(page 10, lines 43 – 44) and because artificially recovering the wetlands in your model has only resulted in a reduction of peak storm surge heights of -2% (page 18 lines 26-28). On the other hand, increasing Manning n coefficients to 0.025 for scattered areas of lost eelgrass resulted in a peak reduction of 3%. This needs some further explanations.*

You are correct in your expectation that recovering fringing wetlands would reduce storm tides significantly and that is shown in the paper (Fig. 8) -- the experiment you are referring to where there was a reduction of only 2% was for "interior" wetlands in the center of the bay, not fringing wetlands. This is explained in the manuscript as being a result of the deep channels running around these wetlands, so that storm tides do not need to pass over them to reach neighborhoods. The experiment restoring fringing wetlands actually reduced the hurricane storm tide more, by 13% (see same section of text). The experiment where bottom roughness was raised to 0.025 across all seabed areas within the bay led to a reduction in the hurricane storm tide of 3%, but I note that this was not just scattered restoration of eelgrass, this was across-the-bay increase in benthic roughness.

The manuscript was modified to be clearer on all these points:
"For example, extensive wetland restoration in the center of the bay (not the fringing wetlands) leads to a change in peak storm tide of only -2%, because deep shipping channels around the wetlands are the primary conduit for flood waters (Orton et al., 2015). A small rise in Mannings-n across the entire bay's seabed from 0.020 to 0.025 (mimicking scattered areas of lost eelgrass, sand bedforms or shells) reduced the peak by -3%. The other changes also had relatively minor effects."

*Page 14, lines 17-20: Please consider moving this sentence to the discussion section, as you start interpreting your results here.*

Change accepted – this text was moved to the end of section 4.1.

*Page 15, lines 25 – 33: This part should be moved into the discussion too.*

This paragraph actually gives results – it is the first time tide range changes are presented. Moreover, the discussion and primary paper focus is on the topic of storm tides. Thus, we have not made this change.

*Page 20, lines 40-41: I suggest rephrasing this sentence to make it a little clearer: It is not the sunlight that is reduced but due to increased turbidity, the light penetration into the water column is reduced.*

Change made, revising to:
"They are known to decline in eutrophic conditions due to the reduced sunlight that results from increased turbidity"

*Figure 4: Perhaps it is just due to the system that created the pdf file, but the legends of both maps are very hard to read. Please check and increase the size of the key.*

Change made, increasing the legend on each panel by 25% in size.

*Literature*
*Lawrence, D.S.L.; Allen, J.R.L.; Havelock, G. M. (2004): Salt Marsh Morphodynamics: an Investigation of Tidal Flows and Marsh Channel Equilibrium. In: Journal of Coastal Research 20 (1); p. 301-316*

*Stark, Jeroen; Plancke, Yves; Ides, Stefaan; Meire, Patrick; Temmerman, Stijn (2016): Coastal flood protection by a combined nature-based and engineering approach. Modeling the effects of marsh geometry and surrounding dikes. In: Estuarine, Coastal and Shelf Science 175; p. 34-54.*

*Stark, J.; Smolders, S.; Meire, P.; Temmerman, S. (2017): Impact of intertidal area characteristics on estuarine tidal hydrodynamics. A modelling study for the Scheldt Estuary. In: Estuarine, Coastal and Shelf Science 198; p. 138-155.*

*Temmerman, Stijn; Vries, Mindert B. de; Bouma, Tjeerd J. (2012): Coastal marsh dieoff and reduced attenuation of coastal floods. A model analysis. Global and Planetary Change 92-93; p. 267-274.*